# Comparative Genome Analysis of 19 *Trueperella pyogenes* Strains Originating from Different Animal Species Reveal a Genetically Diverse Open Pan-Genome

**DOI:** 10.3390/antibiotics12010024

**Published:** 2022-12-24

**Authors:** Zoozeal Thakur, Rajesh Kumar Vaid, Taruna Anand, Bhupendra Nath Tripathi

**Affiliations:** 1Bacteriology Laboratory, National Centre for Veterinary Type Cultures, ICAR-National Research Centre on Equines, Hisar 125001, India; 2Division of Animal Science, Krishi Bhavan, New Delhi 110001, India

**Keywords:** *Trueperella pyogenes*, comparative genome analysis, pan-genome, pyolysin, virulence factors, phylogeny, antimicrobial resistance

## Abstract

*Trueperella pyogenes* is a Gram-positive opportunistic pathogen that causes severe cases of mastitis, metritis, and pneumonia in a wide range of animals, resulting in significant economic losses. Although little is known about the virulence factors involved in the disease pathogenesis, a comprehensive comparative genome analysis of *T. pyogenes* genomes has not been performed till date. Hence, present investigation was carried out to characterize and compare 19 *T. pyogenes* genomes originating in different geographical origins including the draftgenome of the first Indian origin strain *T. pyogenes* Bu5. Additionally, candidate virulence determinants that could be crucial for their pathogenesis were also detected and analyzed by using various bioinformatics tools. The pan-genome calculations revealed an open pan-genome of *T. pyogenes*. In addition, an inventory of virulence related genes, 190 genomic islands, 31 prophage sequences, and 40 antibiotic resistance genes that could play a significant role in organism’s pathogenicity were detected. The core-genome based phylogeny of *T. pyogenes* demonstrates a polyphyletic, host-associated group with a high degree of genomic diversity. The identified core-genome can be further used for screening of drug and vaccine targets. The investigation has provided unique insights into pan-genome, virulome, mobiliome, and resistome of *T. pyogenes* genomes and laid the foundation for future investigations.

## 1. Introduction

*Trueperella pyogenes,* formerly *Arcanobacterium pyogenes*, is a commensal member of skin biota and mucosa of the upper respiratory, gastrointestinal, and urogenital tract of a wide range of animal species including cattle, sheep, and pigs [1]. Infection has also been reported in wild animals [2,3]. This ubiquitously found opportunist bacterium is the etiological agent of abortions and chronic suppurative infections such as mastitis, pyometra, pneumonia, and abscesses leading to significant losses in the livestock industry, especially in intensive systems of animal husbandry [1,4]. The bacterium is mostly observed in multispecies infection withanaerobe *Fusobacterium necrophorum*, which highlights the pathogenic synergy between the two species [5]. Although its importance in animal husbandry is well established, it is being also recognized as playing a role in zoonotic infections in humans [6,7].

The factors that play a pertinent role in the establishment and development of disease includes virulence determinants encoded by pathogens, underlying chronic medical conditions of host, animal husbandry practices, and adverse environmental conditions, among others [1,8].A few virulence determinants associated with the pathogenic potential have been recognized till date such as pyolysin (*plo*), fimbriae (*fimA*, *fimC*, *fimE*,and *fimJ*), collagen-binding protein A (*cbpA*), and neuraminidases (*nanH* and *nanP*) [9,10]. The pyolysin (*plo*) gene belonging to the family of cholesterol-dependent cytolysins is the most recognized and characterized virulence factor of *T. pyogenes*. The haemolytic exotoxin protein encoded by the gene is involved in transmembrane pore formation owing to its cytolytic activity. Notably, *plo* is reported to be present in all of the investigated wild type *T. pyogenes* strains until now [11,12]. Other virulence determinants which targethost tissue and lead to the persistence of *T. pyogenes* include neuraminidase (*nanH* and *nanP*), extracellular binding proteins (*cbpA*), and fimbrial subunits (*fimA*, *fimC*, *fimE*,and *fimJ*) [11,13]. Notably, the knowledge regarding underlying virulence factors involved in the infection development is sparse [1,14]. In order to decode the genetic basis of pathogenesis as well as the unravelling of mechanisms that act as a switch for conversion of a commensal to a pathogen, and comprehensive analysis of *T. pyogenes* strains at genomic level is the first required step.

The first complete genome sequence of *T. pyogenes* species was reported in 2014 of *T. pyogenes* strain 6375 [15]. Till date, 11 complete genome sequences of *T. pyogenes* isolated from various host species and from different countries have been published. In this comparative genomics study of *T. pyogenes* strains from all over the world, we have also included the first Indian draft genome sequence of *T. pyogenes* species (Bu5 strain), which was isolated from wound infections of water buffalo (*Bubalus bubalis*). Notably, infections due to *T. pyogenes* in water buffalo have been reported from Mediterranean and Middle East countries [16,17,18]. In India, cases of *T. pyogenes* infection have been reported from small ruminants [19]; however, reports from buffaloes in India have been elusive.

*Trueperella pyogenes* strains associated with different host species and clinical manifestations have been characterized biochemically, genotypically as well as with different genotype combinations for candidate virulence markers such as *plo*, *fimA*, *fimC*, *fimE*, *fimJ*, *cbpA*, *nanH*, and *nanP* [9,10,20,21]. Pyolysin encoding gene *plo* is detected in all of the pathogenic strains of *T. pyogenes*; however, other candidate virulence determinants such as *nanH*, *nanP*, *cbpA*, and fimbrial genes are not always detected [11,21,22,23]. The high level of variation in biochemical testing and haemolytic reaction on Sheep Blood Agar, resulting in eight different biotypes of *T. pyogenes* isolated from metritis in cattle, also indicates a need for comparative studies between different strains of *T. pyogenes* [22]. However, a comprehensive comparative analysis describing pan-genome and diversity of virulome, mobiliome, as well as resistome of *T. pyogenes* strains has not been reported till date. Althoughfew studies which focussed on fimbrial genes have characterized *T. pyogenes* strain genomes, these studies have utilized a limited number of genes and genomes [9,13,24]. In order to fill the gap, a comparative genome investigation was required to obtain insights of commonalities, dissimilarities, and evolutionary history of the prevalent *T. pyogenes* strains. In fact, identification of new virulence factors significant for pathogenesis could aid in the designing of new strategies for combating of the disease [25,26].

The current study has elucidated pan-genome architecture, which includes core-genome and strain-specific genes in the investigated *T. pyogenes* genomes. In addition, the investigation has also detected diversity in fimbrial genes, candidate virulence genes, genomic islands, prophage sequences, and antibiotic resistance genes via usage of number of computational tools. The identified core-genome can be employed for metabolic profiling as well as for screening of drug or vaccine targets for therapeutic intervention. On the other hand, dissimilar genomic features can be employed as markers for strain characterization, tracking of evolutionary changes, diversity assessment, and determination of transmission history for well-timed detection andepidemiological surveillance. To the best of our knowledge, this is the first comprehensive report of *T. pyogenes* genome analysis encompassing strains from across the world.

## 2. Materials and Methods

### 2.1. Isolation, Characterization, and Genome Sequencing of T. pyogenes Bu5

The *T. pyogenes* (Bu5) strain was isolated from pyogenic wound infection of a water buffalo (*Bubalus bubalis*) from an organized herd at Hisar, (Haryana), India. The isolate was identified by phenotypic colony characteristics, cell-morphology, biochemical tests, and 16S rRNA sequencing. The isolate was grown on Sheep Blood Agar (SBA) at 37 °C in 5% CO_2_ atmosphere. Genomic DNA of *T. pyogenes* Bu5 was extracted using the DNeasy kit (Qiagen), as per the manufacturer’s instructions.

The whole genome sequencing of the Bu5 strain was achieved by shotgun sequencing strategy using 454 pyrosequencing and assembled de novo using Newbler version 2.60. The strain *T. pyogenes* Bu5 (Accession No. VTCCBAA267) is available at the Indian Council of Agricultural Research, National Centre for Veterinary Type Cultures (NCVTC), National Research Centre on Equines (NRCE), Hisar, Haryana, India.

### 2.2. Sequence Information and Quality

The NCBI and the Pathosystems Resource Integration Center (PATRIC) database search for *T. pyogenes* genomes resulted in 24 and 29 entries, respectively, in April, 2021 [27,28]. Nineteen *T. pyogenes* genomes were included in the investigation after exclusion of genomes of low sequence length (GCA_015264715.1, GCA_015264725.1, GCA_015264745.1, GCF_001070855.1, GCA_900299135.1), high number of contigs (GCA_001068695.1), suppressed assembly (GCA_003346995.1), phage genome (GCA_017347915.1), and plasmid sequences (AY255627, U83788) (Appendix A). The sequence information of 19 *T. pyogenes* genomes that were isolated from various parts of the world and consisted of 11 complete and 8 good quality draft genomes including an *in-house* draft genome *T. pyogenes* strain Bu5 were retrieved from NCBI database in fasta format (Appendix A). The investigated genomes have been reported to be isolates of different livestock hosts such as cattle, goats, water buffalo, and pigs (Appendix A). *Trueperella pyogenes* strain TP6375 was utilized as the reference genome in comparative circular plot visualization, pan-genomic analysis as well as in synteny plots. The strain TP6375 was chosen as reference genome in the current investigation due to the following reasons, which include the (i) first complete genome sequence of *T. pyogenes* species, (ii) experimentally utilized for investigation of virulence factors such as fimbrial genes and surface anchored proteins and (iii) also being used as reference genome in the NCBI genome database as well as in previous studies [13,29,30]. The sequence quality information of investigated genomes such as sequencing depth, sequencing platform, assembly method, and other parameters provided by PATRIC is also listed in Appendix A.

### 2.3. Genome Characteristics

The investigated genome sequences were annotated by using the Prokka pipeline available at Usegalaxy server.

(https://usegalaxy.eu/ accessed on 14 April 2021) [31,32]. Basic genomic features, such as the number of CDS, tRNA, rRNA, tmRNA, and repeat regions, were determined.

### 2.4. Pan-Genome, Core-Genome, and Strain-Specific Gene Calculations

Pan-genome describes the complete genetic repertoire of the genomes under investigation, whereas core-genome refers to the set of orthologous genes shared by all the genomes under analysis [33]. On the other hand, strain specific genes or singletons are the unique genes possessed by a genome under study and have no other homologs in the rest of the genomes under analysis [34,35]. In order to determine the pan-genome, core-genome, and strain-specific genes of 19 *T*. *pyogenes* genomes under investigation, EDGAR 3.0 web interface (https://edgar3.computational.bio.uni-giessen.de accessed on 15 July 2021) was employed with default settings [36]. EDGAR is a widely used platform for comparative genome analysis that employs the gene orthology criterion which uses BLAST Score Ratio Values (SRVs) for detection for pan- and core-genome with visual representations. Statistical extrapolation of detected pan- and core-genome is performed by employment of non-linear least-squares curve fitting of the detected core- and pan-genome sizes as a function of the number of investigated genomes in EDGAR [36]. In case of core-genome extrapolation, an exponential decay function as described by Tettelin et al. is used, where *c* is the amplitude of the function, *n* is the number of genomes, Ω is the extrapolated size of the core-genome for *n* → ∞, and τ is the decay constant representing the speed at which *f* converges to Ω [37]:



$$f(n)=c.\;\exp\;\left(\frac{-n}{\tau }\right)+\Omega$$



However, in pan-genome extrapolations, a Heaps’ power law function is employed, where *n* is the number of investigated genomes, *c* is a proportionality constant and γ the growth exponent that depicts at which speed the pan-genome is expanding:*f*(*n*) = *c*⋅*n*^*γ*^

A customized project for pan-genome calculations was created by EDGAR on request.

### 2.5. Functional Annotation

Functional annotation of the representative core-genome subset and complete set of singletons was carried by eggNOG-mapper v2 available at (http://eggnog-mapper.embl.de accessed on 2 December 2022) with default parameters (minimum hit value: 0.001, minimum hit bit score: 60, percent identity: 40, minimum query coverage: 20%, and minimum subject coverage: 20%). The tool utilizes pre-calculated orthologous Groups (OGs) and phylogenies of the EggNOG database (http://eggnog5.embl.de accessed on 2 December 2022) to assign functional annotation to an input dataset of nucleotide sequences. The tool provided COG category assignment, gene ontology (GO), E.C number, KO identifier, and PFAM domain information [38,39].

### 2.6. Average Nucleotide Identity (ANI) Determination

The Average Nucleotide Identity (ANI) is a robust measure of nucleotide-level genomic similarity between two investigated genomes. The estimation of ANI among the investigated *T. pyogenes* genomes as well as the generation of an all versus all comparison matrix was carried at a customized setup at the EDGAR web interface. A clustering tree was also built by utilizing the ANI genome-based distance matrix of the investigated strains [40].

### 2.7. Phylogenetic Tree Construction

Phylogenetic tree was built by utilizing the core-genome of the investigated *T. pyogenes* strains. Each core gene set was aligned by MUSCLE and subsequently the alignments were joined together to form one huge alignment [41]. Finally, the phylogenetic tree was generated from the alignment by using FastTree software for inference of phylogeny. The FastTree method computes local support values calculated by the Shimodaira–Hasegawa (SH) test as metrics of phylogenetic tree reliability [42].

### 2.8. Synteny Plot Analysis

Synteny plots were created by EDGAR for depiction of gene order and detection of large-scale genomic rearrangements if present in the ten investigated complete *T. pyogenes* genomes with reference to *T. pyogenes* 6375 genome [36]. Draft genomes were not included in the synteny plot analysis as their inclusion gives rise to erroneous findings.

### 2.9. Detection of Candidate Virulence Factors (CVFs)

Candidate virulence factors were detected in the investigated 19 *T. pyogenes* genomes by using literature mining, a VFanalyzer tool, BLASTN searches, and manual curation [43,44] as summarized in Figure 1. Firstly, nucleotide sequence information of known virulence genes, namely pyolysin (*plo*), collagen-binding protein A (*cbpA*), neuraminidases (*nanH* and *nanP*), and fimbriae (*fimA*, *fimC*, *fimE,* and *fimJ*) were located and extracted from the genomes of *T. pyogenes* by literature mining and manual curation of the sequence information files available at the NCBI genome database (Appendix A). Next, homologues of these virulent genes were searched for in the rest of the investigated *T. pyogenes* genomes using BLASTN searches. Subsequently, BLASTN search results were then manually checked and, afterwards, corresponding protein sequences were mapped for hit regions in the GenBank or RefSeq assembly sequence records and extracted in fasta format. Multiple sequence alignment of protein sequences of the investigated virulence genes (*plo*, *cbpA*, *nanH*, *nanP*, *fimA*, *fimC*, *fimE,* and *fimJ*) was carried by NCBI COBALT for further inspection [45].

Secondly, the VFanalyzer tool of the VFDB database was employed for the detection of CVFs related to adherence, iron uptake, regulation, toxins, amino-acid and purine metabolism, anti-apoptosis factor, lipid and fatty acid metabolism, phagosome arresting, protease, and stress adaptation among others. The candidate virulent genes that showed differential presence in the investigated genomes by the VFanalyzer tool were searched again using BLASTN searches. In order to maintain stringency, BLASTN results were manually curated and hits that showed query coverage ≥30% and percent identity ≥60% only were considered as homologs.

### 2.10. Detection of Genomic Islands (GIs)

GIs are the cluster of genes of horizontal origin that are attributed to be the source of genetic diversity and contribute to virulence and evolution [46]. Islandviwer4 was used for detection of GIs in the studied *T. pyogenes* genomes. Islandviewer 4 uses four different GI detection methodologies i.e., Islander, SIGI-HMM, Islandpath-DIMOB, and Islandpick [47]. However, GIs in draft genomes were predicted by using SIGI-HMM as well as Islandpath-DIMOB tools by taking *T. pyogenes* 6375 as a reference genome, which was required for contig reordering.

### 2.11. Detection of Prophages

PHASTER (Phage Search Tool Enhanced Release) available at (www.phaster.ca accessed on 17 July 2021) was used to identify and annotate candidate prophage sequences within the *T. pyogenes* genomes. The tool classified identified prophage sequences into the categories of intact, incomplete, and questionable [48].

### 2.12. Searching of Antibiotic Resistance Genes (ARGs)

Searching of ARGs was achieved by employing the CARD database available at (https://card.mcmaster.ca accessed on 22 July 2021). The CARD database archives’ sequences and mutations reported to confer AMR. The resistance gene finder (RGI) tool available at CARD employs detection models on the basis of SNP as well as homology models and subsequently classifies the identified hits in the categories of perfect, strict, and complete category [49].

## 3. Results

### 3.1. Isolation, Characterization, and Sequencing of T. pyogenes Bu5

The pus sample obtained aseptically from an adult water buffalo was streaked on 5% Sheep Blood Agar (SBA) and incubated at 37 °C in the CO_2_ incubator for 24–48 h. After 48 h of incubation, pure, minute haemolytic colonies grew on 5% SBA. *Trueperella pyogenes* strain Bu5 (pus) isolate was Gram-positive cocco-bacilli, non-motile as well as catalase and oxidase negative (Appendix A). A total of 217,058 reads of 424 bp were generated using the GS FLX Titanium system (454 Life Sciences Corporation, Branford, CT, USA), giving ~41× coverage. The data generated 15 large contigs with an average contig size of 148,335 bp and a largest contig size of 741,606 bp. The total size of the genome was 2,225,039 bp, with an N50 of 506,009 bp and a Q40 of 99.95%. This is the first whole genome sequence of *T. pyogenes* isolated from water buffalo (*Bubalus bubalis*) in the world.

### 3.2. Data Availability

This Whole Genome Shotgun project was deposited at DDBJ/ENA/GenBank under the accession PESV00000000, BioprojectPRJNA416992.

### 3.3. Comparative Genome Statistics

Basic genomic features of *T. pyogenes* genomes were determined and compared by using the Prokka pipeline as listed in Table 1. The average genome size of the investigated genomes is 2,327,522.5 bp, ranging from 2,187,257 (*T. pyogenes* DSM 20630) to 2,427,168 (*T. pyogenes* TP4). The average GC% is 59.54% ranging from 59.33% of *T. pyogenes* strain jx18 to 59.8% of the *T. pyogenes* strain MS249. Genomic features such as CDS, rRNA, tRNA, tmRNA, and repeat regions were determined by the Prokka pipeline. The average CDS were observed to be 2079.21 with the highest (2180) in *T. pyogenes* jx18 and the lowest (1948) in *T. pyogenes* Bu5. The tRNA were detected to be in the range of 45–51 with 45 possessed by *T. pyogenes* TP8 and 51 harbored by *T. pyogenes* UFV1. All of the genomes were detected to harbor one transfer-messenger RNA (tmRNA). Repeat regions were observed in the range of 1–10, with two being observed in *T. pyogenes* strains (Bu5, UFV1, and SH01) and the highest number (10) possessed by the *T. pyogenes* strain MS249. On the other hand, the rest of the strains harbored only one repeat region in their genome. The circular plot visualization of other investigated 18 *T. pyogenes* genomes with reference to *T. pyogenes* str. TP6375 depicts varied GC content and GC skew along with core–region similarity (Figure 2).

### 3.4. Pan-Genome Calculations Reveals Open Pan-Genome and Strain-Specific Genes

The pan-genome calculated by the EDGAR 3.0 platform provided us with the entire gene repository of the investigated *T. pyogenes* genomes, which included the core-genome and accessory genome (Figure 3). The pan-genome repertoire of investigated *T. pyogenes* consists of 3214 CDS including a core-genome of 1520 CDS, dispensable genome of 1093 CDS and strain-specific genes in the range of 2–63 (Appendix A). Significantly, a total of 307 CDS in the range of 2–63 were detected as strain-specific genes, also known as singletons in the analyzed genomes (Appendix A). Notably, *T. pyogenes* MS249 harbored the highest number (63) of singletons. There were four strains of *T. pyogenes*, viz., strains TP3, TP6375, TP 4479, and TP 2849 without any unique gene detected in their genomes (Appendix A). The list of CDS identified as components of pan- and core-genome along with strain-specific genes with their name and function are provided in Appendix A.

The pan-genome development plot depicted steady growth with adding up of each new genome and reached 3215 on addition of 19th genome (Figure 3a; Appendix A). However, a core-genome development plot converged to 1488 genes, as with the addition of new genomes, a decrease in shared genes was observed (Figure 3, Appendix A). On the other hand, a singleton development plot depicted that 21 new genes could be found with the adding up of each new genome (Figure 3). The open pan-genome state of *T. pyogenes* species can be inferred by the analysis of pan-genome development plot (growth exponent value of 0.162 (95% confidence interval 0.157 to 0.167)), core-genome development plot, and the singleton genome development plot of investigated strains (Figure 3).

### 3.5. Functional Annotation of Core-Genome and Strain-Specific Gene Repertoire

Functional annotation of representative core-genome subset and complete set of strain-specific genes i.e., singletons, was carried by eggNOG-mapperv2 on the basis of pre-computed orthology assignment. The tool provided functional annotation, orthology assignment, and domain prediction, which included COG category, gene ontology (GO), E.C number, KO identifier, PFAMID, and general description, among others, for each of the analyzed CDS (Appendix A). 1392 CDS (91.57%) out of the total 1520 CDS that were part of the representative core-genome were queried by the Eggnog mapper tool (Appendix A). The maximum number (220) of CDS of core-genome fell into the category of function unknown.

The functional classes that were enriched with the most CDS were carbohydrate metabolism and transport (139), translation (135), inorganic ion transport and metabolism (120), transcription (115), amino acid metabolism and transport (113), energy production and conversion (93), replication and repair (85), and coenzyme metabolism (74), among others (Figure 4A). On the other hand, 97 CDS (31.29%) out of the total 310 CDS of complete subset of singletons were queried by the tool. The repertoire of singletons that was queried by the tool fell into the categories of, replication and repair (23), function unknown (16), defence mechanism (12), transcription (10), and no functional class assigned (20)(Appendix A). A stacked bar graph representing functional roles of the genes unique to individual strain are depicted in Figure 4B and Appendix A.

### 3.6. Phylogenetic and Synteny Plot Analysis

A phylogenetic tree based on 1520 core genes was constructed to decipher the evolutionary relationship amongst the analyzed nineteen *T. pyogenes* genomes. The phylogenetic tree generated depicted three major divergent clades (Figure 5). The Clade I and clade II consist of only large ruminant isolates. Clade I has *T. pyogenes* strains Arash114 and MS249 with the origin from Iran and the UK, respectively. The clade II consisted of Chinese origin cattle strains TP1 and TP2. Notably, the majority (14) of strains were observed in clade III, which bifurcated into two clusters i.e., cattle strains TP6375 (USA) and UFV1 (Brazil), forming one small group and a main large cluster consisting of Asian origin strains (China (11), unknown (2) and India (Bu5)). The Bu5 strain of buffalo origin also makes a separate subgroup with high support, whereas the other subgroup is mainly dominated by the rest of the 10 Chinese origin porcine strains (Figure 5). The SH branch support values of core-genome based phylogenetic tree were very good in general, with only one value of 0.352 below the maximum of 1.00. 

In order to compare the investigated complete genomes against reference strain *T. pyogenes* 6375 and visualize genome scale rearrangements, synteny plots were constructed by utilizing the EDGAR web interface. The synteny plots depicted synteny and large-scale genomic rearrangements such as inversion, duplication, relocation, and palindrome (Figure 6). Notably, the 2012CQ-ZSH genome showed the highest collinearity with the reference genome *T. pyogenes* 6375. Large inversion events were observed in the synteny visualization of 2012CQ-ZSH, TP3, TP8, TP2849, and TP 4479 against the reference genome (Appendix A). In addition, relocation and duplication were also observed in the synteny plots of TP3 and TP8 genomes against the reference genome (Appendix A). In comparison, smaller inversions along with deletions and duplications were observed in the synteny plots of jx18, TP1, and TP4 against the reference strain (Appendix A). In particular, a large palindrome with duplication along with high synteny was also observed in the TP2 synteny plot against the reference genome (Appendix A). The synteny plots of each of the analyzed complete genomes against the reference genome *T. pyogenes* 6375 are depicted in Appendix A.

### 3.7. Decoding of Virulome: Candidate Virulence Genes Identified

An inventory of potential virulence factors was identified by the combinatorial usage of the VFanalyzer tool, BLASTN searches, literature mining, and manual curation in the investigated *T. pyogenes* genomes.

Firstly, we searched and analyzed homologues of principal virulence factors pyolysin (*plo*) along with other putative virulence factors such as neuraminidases (*nan*Hand *nan*P), extracellular matrix-binding proteins (*cbpA*), and fimbriae (*fimA*, *fimC*, *fimE*, and *fimJ*), which are involved in adherence and colonization of the host tissue in the studied *T. pyogenes* strains by employing literature mining, BLASTN searches, and manual curation. The information related to BLASTN search results of detected homologues, which includes reference sequence used, query coverage, percent identity, start position, and end position, is tabulated in Appendix A. The homologs of cholesterol dependent pyolysin gene (*plo*) of *T. pyogenes* 6375 were observed to be part of the core-genome of all of the investigated *T. pyogenes* genomes with a relatively high sequence identity in the range of 80.8% in Bu5 to 100% in UFV1 (Appendix A).

On the other hand, homologues of *T. pyogenes cbpA* were located and analyzed in all of the investigated *T. pyogenes* strains and that varied in the range of 53% QC and sequence similarity 97.66% in the SH01 strain to 100% QC and sequence similarity 96.21% in TP1 (Appendix A). The *cbpA* homologues in strains such as TP2, MS249, UFV1, and SH01 were observed to be truncated as depicted by the multiple sequence alignment of their corresponding protein sequences. (Appendix A). Gene truncation at N terminal was observed in strains MS249, UFV1, and SH01, whilst C terminal truncation was observed in *cbpA* of TP2.

Similarly, homologues of *T. pyogenes nanH* were identified in all of the studied genomes with varying sequence identity in the range of 100% QC and sequence similarity 87.62% in TP2 to 78% QC and 87.59% sequence similarity in UFV1. Multiple sequence alignment of protein homologues of *nanH* depicted missing N terminal fragments in TP6375, jx18, DSM20630, SH01, SH03, 2012CQ-ZSH, and UFV1, and a missing C terminal fragment with gaps in reference protein, TP2, TP8, MS249, Bu5, and NCTC5224 (Appendix A).However, both N terminal and C terminal fragments were observed to be missing in the UFV1 strain. On the other hand, *T. pyogenes nanP* homologues were identified in only 12 out of the 19 investigated *T. pyogenes* genomes with varying sequence identity in the range of 100% QC and 99.37% sequence similarity in Bu5 to 55% QC and 98.62% sequence similarity in MS249. Multiple sequence alignment depicted *nanP* homologues to be highly conserved with the exception of missing N terminal and C terminal fragment in TP8 and missing C terminal fragment in MS249 (Appendix A).

Next, we searched the homologues of TP6375 fimbrial genes i.e., *fimA*, *fimC*, *fimE*, and *fimJ* in the *T. pyogenes* genomes. Amongst the four TP6375 fimbrial genes investigated, *fimC* and *fimE* were detected to be highly conserved in all of the investigated strains. The homologues of TP6375 *fimA* were observed to be highly conserved in most of the investigated *T. pyogenes* strains, with the exception of few strains wherein truncation of *fimA* was observed, which includes strains 2012CQ-ZSH, DSM20630, NCTC5224, and UFV1 (Appendix A). The homologues of TP6375 *fimA* varied in the range of QC 21% and sequence similarity 95.92 in DSM20630 and NCTC524 to QC 100% and sequence similarity 98.99% in TP1 (Appendix A). Truncation of C terminal fragment along with a gap was observed in the *fimA* of TP6375 and UFV1 and truncation of N terminal fragment was found in NCTC5224, DSM20630, and 2012CQ-ZSH in the multiple sequence alignment of *fimA* protein homologues (Appendix A). Notably, homologues of TP6375 *fimC* were observed to be conserved in all of the investigated *T. pyogenes* genomes with sequence identity ranging from QC 99% and 70.01% sequence similarity in TP3, TP4479, and TP2849 to 100% QC and 100% sequence similarity in UFV1 (Appendix A). Similarly, homologues of *fimE* TP6375 were observed in all of the investigated *T. pyogenes* genomes with sequence coverage and sequence identity in the range of sequence coverage of 87% and sequence similarity of 74.2% in MS249 to sequence coverage and sequence similarity of 100% in UFV1 (Appendix A). On the contrary, homologues of TP6375 *fimJ* were observed in all of the investigated genomes with the exception of UFV1 and Bu5. The homologues varied in the range of query coverage 17% and sequence similarity 99.26% in SH01 and 2012CQ-ZSH to sequence coverage 100% and sequence identity 99.06% in Arash114 (Appendix A).

Apart from searching and analysis of homologues of *plo*, *fimA*, *fimC*, *fimE*, *fimJ*, *cbpA*, *nanH*, and *nanP,* we also utilized the VFanalyzer tool to search other candidate virulence genes (Figure 1). The VFanalyzer tool of VFDB along with BLASTN searches and manual curation detected 500 potential virulence genes that were observed to be homologues of known virulence genes of bacterial pathogens belonging to genuses such as *Mycobacterium*, *Klebsiella*, *Haemophilus*, and *Francisella*, among others, as detailed in Appendix A. The potential virulence factors identified in the study appear to be associated with a wide range of virulence related functions such as adherence, iron uptake, regulation, toxin, and amino-acid and purine metabolism, anti-apoptosis factor, lipid and fatty acid metabolism, phagosome arresting, biofilm formation, protease, and stress adaptation, among others (Figure 7 and Figure 8, Appendix A). The current investigation revealed the presence ofhomologues of genes related to iron uptake and the siderophore biosynthesis system i.e., *ciu*A, *ciu*B, *ciu*C, and *ciu*D along with a homolog of ABC-type heme transporter related gene *hmu*U in all of the investigated *T. pyogenes* genomes. Additionally, homologues of mycobacterial regulatory proteins such as *rel*A, *reg*X3, *sig*H, and *sig*A/*rpo*Vwere also observed in each of the investigated genomes. Similarly, homologues of mycobacterial genes related to lysine synthesis (*lys*A), glutamine synthesis (*gln*A1), anti-apoptosis factor (*nuo*G), protease (*mpa*, *zmp*1), stress adaptation (*sod*A), and anti-phagocytosis (*rml*A) were also found by the usage of a VFanalyzer tool, NCBI BLASTN searches, and manual curation in all of the genomes under study. In a similar manner, homologs of genes related to the secretion system (T6SS-II from *Klebsiella*), capsule (*rml*Bfrom *Streptococcus*), and pyrimidine biosynthesis (*Francisella*) were also detected in all of the investigated genomes. On the contrary, genes related to adherence (*srt*B), ABC transporter (*fag*C), ABC-type heme transporter (*hmu*V), regulation (*pho*P, *whi*B3), lysine synthesis (*lys*A), pantothenate synthesis (*pan*C, *pan*D), capsule (*gnd*), trehalose-recycling ABC transporter (*sug*C), exopolysaccharide (*gal*E), pyrimidine biosynthesis (*car*B), and exopolysaccharide (*gal*E and *pgi*) showed differential distribution in the investigated genomes (Figure 7 and Figure 8, Appendix A).

### 3.8. Genomic Island Detection

A total of 206 GIs were detected in the investigated *T. pyogenes* genomes. The detected genomic islands varied in the range of 14–25 and 4.00–82.09 kb in terms of number and size, respectively. The highest numbers (25) of GIs were found in *T. pyogenes* SH02, and the largest sized GI (82.093 kb) was observed in *T. pyogenes* Arash114. However, the lowest number (12) of GIs were detected in *T. pyogenes* TP8 and the smallest size (4 kb) GI in *T. pyogenes* TP2 (Figure 9a). Data related to identified GI’s such as starting position, end position, and size are listed in Appendix A.

### 3.9. Prophage Detection

A total of 30 prophage sequences were detected in all of the investigated *T. pyogenes* genomes in which two were classified as intact, 26 as incomplete and two as questionable by PHASTER (Appendix A). The identified prophage sequences varied in the range of 1–4 and 5.2–47.7 kb in terms of number and size, respectively, in the investigated genomes. The GC% of identified prophages varied in the range of 52.20−64.49%. The average size of identified prophages is 19.79 kb. The highest number (4) of prophage sequences were found in the *T. pyogenes* TP1 genome in which three were classified as incomplete prophage sequences and one as an intact prophage sequence. Only one prophage sequence was detected in the genomes of *T. pyogenes* strains 2012CQ-ZSH, TP2, TP8, DSM 20630, MS249, SH01, SH02, SH03, UFV1, and NCTC5224 by PHASTER, and all of them were classified in incomplete categories (Figure 9b). The detailed information of each identified prophage sequence such as region length, completeness, score, region position, most common phage and GC% are listed in Appendix A.

### 3.10. Antibiotic Resistance Genes (ARG) Detection

A total of 40 ARGs were detected in the investigated genomes by CARD in which 35 were classified into strict and five into perfect category according to RGI criteria (Figure 10). The ARGs which were detected mainly conferred resistance against aminoglycosides (*APH*(3’)-Ia, *APH*(6)-Id, *rmt*B, *ANT*(3’’)-Ia, *ANT*(2’’)-Ia) tetracyclines (*tet*(W/N/W)*, Tet*Z, *Tet*33), phenicols (*cml*A6), sulphonamides (*sul*1), and streptogramin, macrolides, and lincosamide antibiotics (*erm*X) apart from disinfectants and antiseptics (*qac*E delta1).

The highest numbers of ARGs were found in *T. pyogenes* SH01 (6), *T. pyogenes* SH02 (6), and *T. pyogenes* TP1 (5), respectively. Notably, no ARG were detected in genomes of *T. pyogenes* strains DSM20630, NCTC5224, Bu5, and UFV1 by CARD (Appendix A).

## 4. Discussion

*Trueperella pyogenes* is a resident commensal of skin biota and mucous membrane of the upper respiratory and genital tracts of domestic and wild animals including humans [1]. The bacterium has also been detected in bovine rumen and the gastrointestinal tract of swine [50]. However, in response to a precipitating injury or infection, the bacterium manifests as one of the most common opportunistic pathogens, resulting in a range of purulent infections such as mastitis, cutaneous and liver abscessation, metritis, endometritis, and pneumonia in domestic and wild animals, which results in huge economic losses [1,11,51,52,53]. Its versatility can be gauged from the range of animals it has been isolated from, including wild ungulates (antelopes, bison, deer, musk deer, gazelles, wildebeest), avian species (chicken, macaws, turkeys) elephants, reindeer, and companion animals (dogs, cats, and horses). Even though the species has been recognized for a long time, still the underlying mechanisms of disease pathogenesis, reservoirs as well as routes of transmission of bacteria, are incompletely understood [1]. The rise in drug resistance to available antibiotics poses a significant challenge to control the disease effectively [23,52,54]. Vaccination can be an important and effective measure for the control of pathogen, but no commercial vaccine with adequate protection is available till date [55,56,57,58]. Consequently, the understanding of genomic architecture of prevalent *T. pyogenes* strains is crucial for development of the strategies to control the thriving of pathogenic strains. Significantly, only a few virulence factors, namely, pyolysin(*plo*), fimbriae (*fimA*, *fimC*, *fimE*, and *fimJ*), collagen-binding protein A (*cbpA*), and neuraminidases (*NanH* and *NanP*),which contribute to the pathogenic potential, have been recognized [1,11,13]. However, the complete role of such virulence factors can only be understood after elucidation of core-genomic features of this pathogen.

In the current investigation, we have compared all the available 19 *T. pyogenes* strains originating from distinct geographical regions i.e., China (11), India (1), Iran (1), US (1), Brazil (1), and Australia (1) to elucidate the gene repertoire as well as their distinct genomic features (Appendix A). The study encompasses eleven completely sequenced genomes and eight good quality draft genomes isolated from different host species (Appendix A). In order to obtain insight into the genetic repertoire of the analyzed strains, pan-genome calculations were performed. The pan-genome investigation of *T. pyogenes* genomes revealed a pan-genome repertoire of 3214 CDS, a core-genome of 1520 CDS (47.3%), a dispensable genome of 1093 CDS (34%), and strain-specific genes in the range of 2–63 (18.7%), respectively (Figure 3d). The core-genome, which is nearly 47.3% of the pan-genome, reveals that a high level of intra-species diversity exists at the genomic level among *T. pyogenes* strains included in the study. The pliant genomic subset comprised of dispensable genome and singletons is nearly 52.7% of the total pan-genome in this study, which indicates the capacity and propensity of *T. pyogenes* to adapt to the challenges posed by a variety of warm-blooded host cell surface and environmental stressors such as antimicrobial compounds. Notably, no strain-specific genes were observed in the genomes of TP3, TP6375, TP4479, and TP2849; however, it is notable that 3 strains out of 4, i.e., TP3, TP4479 and TP2849, are almost identical genomically with average nucleotide identity (ANI) of 100% (Appendix A). The fewer number of singletons detected in our study has similarly been observed in pan-genome analysis of 42 *Arcanobacterium phocae* strains which contained 73 unique genes [59]. The *Arcanobacterium* spp. and *Trueperella* spp. are phylogenetically close taxa having been recently separated [60].

The pan-genome development graph of investigated *T. pyogenes* strains depicts steady growth with the addition of each genome, which reflects the genomic diversity of the investigated strains as well as the capacity of *T. pyogenes* to acquire exogenous DNA (Figure 3). The core-genome development plot converges to about 1488 genes (Figure 3). This set of genes shared by all analyzed strains play an important role in bacterial survival [33,35,37] and in the case of *T. pyogenes*, this gene set may be helpful in defining its unique ability of a commensal with an ability to cause opportunistic pyogenic infections in a wide variety of mammals. The singleton development plot depicted the possibility of finding 21 new genes with the addition of a new *T. pyogenes* genome (Figure 3). Notably, the steady growth in the pan-genome development plot with a growth exponent value of 0.162, convergence in the core-genome development plot, and the possibility of finding ≈21 new genes, with the addition of a newly sequenced genome, led to significant inference that *T. pyogenes* genomes harbor an open pan-genome state (Appendix A, Figure 3). However, the number of genomes used in our study is limited to 19, which may be a limiting factor in true estimation of genome openness. For example, *A. baumannii* strains were estimated to be closed by Chan et al. (2015), when they took 249 genomes into account, which was previously estimated to be open on analysis of the pan-genome size of 16 strains [61,62]. However, two observations make our conclusion of pan-genome open status as significant, i.e., one in which 19 *T. pyogenes* strains have a wide origin on the basis of geography, host, and core-genome phylogeny (Table 1; Figure 5), and secondly, the ANI value of all 19 strains have been calculated to be ≥97.5% (Appendix A) [63].

Next, we carried functional annotation of representative core-genome subset and a complete subset of strain-specific genes of investigated *T. pyogenes* genomes by utilizing an eggNOG-mapperv2tool. The tool provided functional annotation, orthology assignment, and domain assignment to the input genomic subsets. Notably, 91.57% CDS of the core-genome subset and 31.59% CDS of strain-specific genes were assigned into different COG functional classes by the eggNOG-mapperv2. The maximum CDS (220) of core-genome subset fell into the function unknown category. Next, the functional classes that were enriched with most CDS of core-genome after the function unknown category includes carbohydrate metabolism and transport (139), translation (135), inorganic ion transport and metabolism (120), transcription (115), amino acid metabolism and transport (113), and energy production and conversion (93), replication and repair (85), defence mechanism (31), among others (Appendix A; Figure 4A). On the other hand, 31.29% CDS of strain-specific genes subset were assigned into functional categories of replication and repair (23), function unknown (16), defence mechanism (12), and transcription (10), among others (Appendix A; Figure 4B). The functional analysis underlines the metabolic versatility of *T. pyogenes*, as it encompasses most of the basic categories of gene ontology [64].

In order to assess the evolutionary relationship among the investigated TP genomes, core-genome was utilized for phylogenetic tree construction. The strains of *T. pyogenes* were found to be falling into different phylogenetic clusters with statistically significant high SH branch support values (Figure 5). The phylogenetic tree built on the basis of 1520 core genes depicted three distinct clades, wherein clade I comprised of strains Arash114 (Iran) and MS249 (Australia) and clade II consisted of Chinese origin strains TP1 and TP2. Notably, clade III harbored the maximum number of strains (15) which bifurcated into a small group consisting of TP6375 (USA) and UFV1 (Brazil) and a large subclade comprising of Asian origin (India- 1, China- 11, and unknown origin-2) strains (Figure 5). The phylogenetic tree gives a picture of high degree of genomic diversity and evolution within the core-genome level of species, which agrees with the propensity of *T. pyogenes* to cause opportunistic infections in a large variety of warm-blooded animals [4].

Interestingly, the three clades can be divided into two groups based on the animal host associated with the strain. Clades I and II consist of domestic large ruminants including cattle and buffalo, whereas clade III is almost entirely derived from the porcine host. Although, within clade III, extensive branching has divided the strains into multiple subgroups with high SH values, the strain Bu5, which is an Indian water buffalo origin strain, is forming a separate clade (Figure 5). The core-genome phylogeny assignment of *A. phocae*, which is a close relative of *T. pyogenes*, also divided the 42 investigated strains into three clusters, with the different clusters predominated by different hosts [59]. The genetic variation in *Escherichia coli* among strains of different phylogroups is believed to support fitness in different ecological habitats, leading to niche preference [65].The benefit of strains identity at clonal or phylogroup level has also been underlined as a function of its environmental niche, mode of living, and ability to cause disease [66,67]. The core-genome phylogeny in the current investigation shows clear distinction between the ancestral ruminant strains and porcine strains. However, in order to draw better inferences of evolutionary history, the addition of more completely sequenced genomes of *T. pyogenes* isolated from various hosts and geographical regions is required. Additionally, to understand the genomic architecture and possible evolutionary events in *T. pyogenes* strains, synteny plots were created with a reference genome taken as *T. pyogenes* 6375. The synteny plot of investigated complete genomes depicted high synteny with large scale genomic rearrangements such as inversions, insertion, deletion, and relocations (Figure 6; Appendix A). In particular, 2012CQ-ZSH genome showed a high degree of synteny conservation with the reference genome *T. pyogenes* 6375 (Appendix A).

Although reports of isolation of *T. pyogenes* from bovine rumen, and gastrointestinal tract of swine are there [50], the ecological niche of *T. pyogenes* is considered to be a mucus membrane and skin wherein the bacterium gains entry as an opportunistic pathogen after violence in the mucous membrane of the respiratory and genital tract [68,69]. Apart from this property, the synergistic collaboration of *T. pyogenes*, in conjunction with bacteria such as with anaerobe *Fusobacterium necrophorum* in inter digital pyogenic infections, and with *E. coli* in urogenital infections, shows the strategy of pathogenic synergy with many different bacteria [5]. *Trueperella pyogenes* is a highly versatile and adaptable pathogen, as it is able to produce many factors which aid its adherence and colonization in different body locations, both internally and externally. Therefore, decoding of virulence factors associated with *T. pyogenes* is crucial for the understanding of molecular pathogenesis and development of therapeutics for prevention and control [70,71]. In the current investigation, we utilized BLASTN searches, the VFanalyzer tool, and manual curation to identify and analyze putative virulence genes in the investigated *T. pyogenes* genomes. First, we utilized BLASTN searches, and manual curation to identify and analyze putative virulence genes which includes pyolysin (*plo*), collagen-binding protein A (*cbpA*), neuraminidases (*NanH* and *NanP*), and fimbriae (*fimA*, *fimC*, *fimE*,and *fimJ*) in the investigated *T. pyogenes* genomes. BLASTN searches revealed the *plo* gene to be a highly conserved component of the core-genome, as homologs of *T. pyogenes* 6375 *plo* were observed in all of the investigated *T. pyogenes* strains with high sequence identity ranging from 80.8% (Bu5) to 100% (UFV1) (Appendix A). Multiple sequence alignment of the *plo* homologues suggested high sequence identity with gaps being observed only at two residues (alignment position 17th harboring glycine and 58th possessing threonine) in the Bu5 *plo* sequence (Appendix A). In corroboration with our results, the presence of *plo* in all of the investigated wild type *T. pyogenes* strains has been previously reported [9,10,21]. plo is considered to be the sole haemolysin and crucial for *T. pyogenes* survival, which makes it a highly attractive drug and vaccine target. A number of attempts have been previously made for exploitation of *plo* gene as a vaccine target by utilizing different strategies but has failed to provide adequate protection against the lethal pathogen [57,72]. Therefore, new strategies involving novel targets or altered approaches to utilize *plo* are vital to counter the pathogen.

Adhesion to epithelial cells is generally the first step in pathogenesis and is crucial for the ability of bacteria to colonize host mucosal surfaces [73,74,75,76]. Therefore, virulence factors associated with adhesion and immunological response such as neuraminidase (*nanH* and *nanP*)*,* extracellular binding proteins (*cpbA*)*,* and fimbriae (*fimA*, *fimC*, *fimE*, and *fimJ*) are considered significant for the establishment of *T. pyogenes* infection inside the host [1,11]. Therefore, next, we located and analyzed these potential virulence factors in the *T. pyogenes* genomes. *Trueperella pyogenes nanH* homologs were detected in all of the investigated genomes with varying sequence identity ranging from 100% QC and sequence similarity 87.62% in TP2 to 78% QC and 87.59% sequence similarity in UFV1 (Appendix A). The multiple sequence alignment of nanH protein homologues described truncated N-terminus fragments in the strains TP6375, jx18, DSM20630, SH01, SH03, 2012CQ-ZSH, as well as in UFV1 and truncated C-terminus fragments with gaps in reference protein in the strains TP2, TP8, MS249, Bu5, and NCTC5224, whereas both N-terminus and C-terminus fragments were detected to be truncated in UFV1 strains (Appendix A). On the other hand, *T. pyogenes nanP* homologues were identified in only 12 out of the 19 investigated *T. pyogenes* genomes with varying sequence identity in the range of 100% QC and 99.37% sequence similarity in Bu5 to 55% QC and 98.62% sequence similarity in MS249 (Appendix A). The multiple sequence alignment of nanP protein homologues was observed to be highly conserved with the exemption of truncated amino and carboxy terminal fragments in TP8 and truncated carboxy terminal fragments in MS249 (Appendix A). Neuraminidases (*nanH* and *nanP*) promote host cell adhesion by cleaving off terminal sialic acids and thus exposing host cell receptor molecules. Neuraminidase also decreases mucous viscosity, which facilitates bacterial colonization in underlying tissues. Moreover, the enzyme has been reported to impair the host immune response as susceptibility of mucosal IgA to bacterial proteases is increased [1,11].

Additionally, *T. pyogenes* also harbors virulence proteins that aid in binding to extracellular matrix binding proteins such as collagen, fibrinogen, and fibronectin. However, only collagen binding ability and not fibrinogen and fibronectin binding ability has been characterized in *T. pyogenes*. CbpA, an MSCRAMM-like surface protein, exploits collagen types I, II, and IV for adherence and subsequent colonization in collagen rich tissue. CbpA is comprised of signal peptide, collagen binding domain, repetitive B domains, and a cell wall anchoring domain. Reduced binding to epithelial and fibroblast cell lines has been demonstrated in *cbpA* knockout mutant [1,11]. The homologues of *T. pyogenes cbpA* were positioned and analyzed in all of the investigated *T. pyogenes* strains and that showed variation in the range of (53% QC and sequence similarity 97.66%) in SH01 to (100% QC and sequence similarity 96.21%) in TP1 Appendix A. The *cbpA* homologues in strains such as TP2, MS249, UFV1, and SH01 were observed to be truncated as depicted by the multiple sequence alignment of their corresponding protein sequences (Appendix A). Gene truncation at N terminus was observed in strains MS249, UFV1, and SH01, while C terminal truncation was observed in *cbp*A of TP2.

TP6375 *fimA* homologues displayed a high degree of sequence identity in all of the investigated genomes, with the exception of a few strains wherein gene truncation of *fimA*was observed, which includes strains 2012CQ-ZSH, DSM20630, NCTC5224, and UFV1 (Appendix A). Consequently, multiple sequence alignment of *fimA* homologues depicted a high degree of conservation with the exception of *fimA*of 2012CQ-ZSH, DSM20630, and NCTC5224, which displayed missing N terminal fragments and of UFV1 that displayed missing C terminal fragments. Homologues of *fimC*of TP6375 in all of the investigated *T. pyogenes* genomes were observed to be conserved with a sequence identity in the range of 100% in UFV1 to 70.01% in TP3, TP4479, and TP2849 (Appendix A). Similarly, homologues of *fimE* TP6375 were observed in all of the investigated *T. pyogenes* genomes with percent identity in the range of 98.23% in Arash114 to 68.55% in NCTC5224 and DSM20630. It is worth notingthat homologues of TP6375 *fimJ* were observed in all of the investigated *T. pyogenes* strains with the exception of Bu5 and UFV1 strains in the range of 100% query coverage and sequence identity of 99.11% in Arash 114 to 17% query coverage and sequence identity of 99.26% in SH01 strains (Appendix A). The multiple sequence alignment of fimbrial protein sequences revealed that *fimC* and *fimE* harbored lesser mutation events in comparison with *fimA* and *fimJ* in the analyzed sequences (Appendix A). The expression profile of putative fimbrial proteins i.e., *fimA*, *fimC*, and *fimE*, in cultured *T. pyogenes,* was determined by Liu et al., (2018) by cloning respective fimbrial proteins and then generating rabbit anti-rFimA, anti-rFim C, and anti-rFim E serum. The Western blot assay performed using these sera revealed that only *fim*E was constitutively expressed in *T. pyogenes* [24].

Out of all the investigated candidate virulence factors, *plo* appears to be strongly conserved among the investigated genomes. Risseti et al. (2017) investigated 71 *T. pyogenes* strains recovered from mastitis (*n* =35), and non-mastitis and reported the presence of *plo* (100.00%), *fimA* (98.6%), *nanP* (78.9%), *fimE* (74.6%), *fimC* (64.8%), *nanH* (63.4%), *cbpA* (8.4%) and *fimG* (5.6%) in their studied strains [9].The variability in gene content is the basis of bacterial evolution, and gene truncation is significant for shaping bacterial genomes [77]. The truncated candidate virulence genes detected in the study (Appendix A) might be the result of small-scale lateral gene transfer events and could possibly encode shortened proteins with different functions which can be unravelled by experimental investigation [77,78]. On the other hand, the indels detected in the current study could also be present for reverse gene silencing i.e., for adaptations with respect to changing conditions (Appendix A). The indel frequency inversely correlates with linguistic complexity of genome, gene–position as well as gene–essentiality [79].

After careful and comprehensive analysis of homologues of *plo*, *fimA*, *fimC*, *fimE*, *fimJ*, *cbpA*, *nanH*, and *nanP*, we applied the VFanalyzer tool and manual curation, which led to identification of 500 candidate virulence genes (Figure 7 and Figure 8). The identified virulence genes shared homology with known virulence genes of pathogens belonging to genera such as *Mycobacterium*, *Klebsiella*, *Haemophilus*, and *Francisella*, among others, as detailed in Appendix A. The candidate repertoire of virulence factors identified in the study seems to be associated with a wide range of virulence related functions such as adherence, iron uptake, biofilm formation, regulation, toxin, amino-acid and purine metabolism, anti-apoptosis factor, lipid and fatty acid metabolism, phagosome arresting, protease, and stress adaptation as listed in Appendix A. Notably, the homologues of genes related to iron uptake and siderophore biosynthesis system i.e., *ciu*A, *ciu*B, *ciu*C, and *ciu*D along with a homolog of ABC-type heme transporter related gene *hmu*U were present in all of the investigated *T. pyogenes* genomes. The bacterial systems for iron uptake, such as high-affinity siderophore uptake systems, are important virulence factors in many Gram-positive bacterial pathogens such as *C. diptheriae* and *C. pseudotuberculosis* [80,81]. In addition, homologs of mycobacterial regulatory genes, such as *relA, regX*3, *sigH*, and *sigA/rpoV*, were also detected to be part of the core-genome of all of the investigated *T. pyogenes* genomes. These mycobacterial regulatory genes play a pivotal role in bacterial persistence (*relA*), nutrient sensing (*regX*3), encountering of thermal and oxidative stress (*sig*H), and growth promotion in macrophages (*sigA/rpoV*) [82,83,84,85].

Similarly, homologues of mycobacterial genes related to lysine synthesis (*lysA*), glutamine synthesis (*glnA*1), anti-apoptosis factor (*nuoG*), protease (*mpa, zmp*1), stress adaptation (*sodA*), and antiphagocytosis (*rmlA*) were also found by the usage of the VFanalyzer tool, NCBI BLASTN searches, and manual curation in all of the genomes under study. In a similar manner, homologs of *Klebsiella* T6SS-II, mycobacterial GPL locus (*rml*A), and streptococcal capsule (*rmlB*) were also identified in all of the studied genomes.

In contrast, genes related to adherence (*srtB*), ABC transporter (*fagC*), ABC-type heme transporter (*hmuV*), regulation (*phoP*, *whiB*3), lysine synthesis (*lysA)*, pantothenate synthesis (*panC*, *panD*), capsule (*gnd*), trehalose-recycling ABC transporter (*sugC*), exopolysaccharide (*galE*), pyrimidine biosynthesis (*carB*), and exopolysaccharide (*galE and pgi*) showed differential distribution in the investigated genomes (Appendix A).

Microbial genomes are dynamic in nature as they evolve and adapt with time through horizontal gene transfer (HGT), mutations, and gene rearrangements [86,87]. These adaptations aid in better survival of bacterium in response to stresses encountered in varying conditions [88,89]. As mobile genetic elements (MGEs) add to the resilience of genome, we next searched for MGEs which include genomic islands, prophages, and antibiotic resistance genes by employing various bioinformatics tools in the investigated *T. pyogenes* genomes.

Genomic islands are the cluster of genes attained by HGT and known to be significant for pathogenesis, fitness, and antibiotic resistance [90]. A total of 346 genomic islands in the range of 12–25 in terms of number were found in the investigated *T. pyogenes* genomes by Islandviewer (Appendix A, Figure 9a). Highest and lowest numbers of GIs were detected in the strains SH02 and TP8, which comprise 16.5% and 2% of their genome, respectively, by Islandviewer. Genomic islands of *T. pyogenes* strain TP3 (TP3-GI1 and TP3-GI5) and *T. pyogenes* strain TP4 (TP4-GI5 and TP4-GI8), which are involved in multidrug resistance, were investigated by Dong et al., 2020. The investigators documented that TP3-GI1 and TP4-GI5 shared a common region of 20 kb, which is comprised of tetracycline resistance gene tet(W) along with a series of genes involved in type IV secretion systems [91]. On the other hand, TP3-GI5 and TP4-GI8 also shared homology, which includes a series of phage and DNA replication linked genes. The TP3-GI5 is comprised of two IS6100 located in the same orientation and one tet(33) forming a composite transposon along with macrolide resistance gene erm(X) located near the end, whereas TP4-GI8 harbors two copies of erm(X) present between two IS1634 elements placed in the same orientation which might have formed a composite transposon. Except for these four GIs of TP3 and TP4 strains, the structural composition and role of other GIs in infection pathogenesis have not been investigated yet. In the current investigation, we have not further attempted to analyze the identified GIs, as it is beyond the scope of our study.

Bacteriophages are the most abundant genetic entities on the biosphere that influence bacterial virulence and evolution in a number of ways [92]. The lytic reproduction and temperate reproduction (prophage) of bacteriophage depict the extremes of parasitism and mutualism within a single phage genotype [93]. Prophages are significant genetic elements of bacterial genomes as they contribute to diversity, virulence, and fitness [94]. The current study identified 31 prophages in the range of 1–4 in terms of number and 5.2–47.7 kb in terms of size. Out of 31 prophages, two were classified as intact, two as questionable and 27 as incomplete (Appendix A; Figure 9b). The average prophage number was detected to be 1.63 per genome. The two intact prophage elements which shared structural similarities with *Lactobacillus* phage iA2 (NC_028830) and *Staphylococcus* phage SPbeta-like (NC_029119) were observed in the strains *T. pyogenes* TP6375 and *T. pyogenes* TP1, respectively. However, the maximum number (4) of prophage elements was detected in the strain TP1 in which one was classified as complete and three as incomplete prophage elements. The most abundant phage type was found to be “PHAGE_Lactob_iA2”, “PHAGE_Coryne_Adelaide”, and “PHAGE_Salmon_118970_sal3” with each type being found in four investigated *T. pyogenes* strains. Interestingly, seven out of the 31 identified prophage elements were unique i.e., were not carried by any another analyzed *T. pyogenes* strains, which include PHAGE_Propio_E6_NC_041894 (jx18), PHAGE_Staphy_SPbeta_like_NC_029119 (TP1), PHAGE_Mycoba_Gaia_NC_026590 (TP4), PHAGE_Coryne_Stiles_NC_048789 (TP8), PHAGE_Synech_S_CAM8_NC_021530 (MS249), PHAGE_Coryne_Lederberg_NC_048790 (NCTC5224), and PHAGE_Brevib_Jimmer2_NC_041976 (Bu5). The unique prophage repertoire detected in the study depicts that the strains harboring them have encountered different environmental conditions and thus have acquired unique prophage sequences. The diverse prophage repertoire is maintained by evolutionary forces such as positive selection of host beneficial genes, negative selection of lytic function genes, and mutational bias of bacteria towards deletion leading to gene loss [93].

Resistome analysis of genomes via CARD database revealed 40 Antibiotic resistance genes (ARGs) wherein five were classified in perfect and 35 in the strict category (Appendix A; Figure 10). The identified ARGs in the current study are reported to show resistance to antibiotics such as macrolide, lincosamide, streptogramin, and tetracycline. Highest numbers (6) of ARGs were detected in the Chinese origin strains *T. pyogenes* SH01 and SH02 and that exhibited resistance to antibiotics—aminoglycoside, tetracycline, phenicol, sulphonamide, and acridine dye. Notably, 13 out of the 19 investigated genomes harboured the *tet* (W/N/W) gene which encodes for mosaic tetracycline-resistant ribosomal protection protein. Literature mining also revealed the resistance of *T. pyogenes* to tetracycline in dairy cows suffering from endometritis in Inner Mongolia and China [95]. Recently, 114 *T. pyogenes* isolates obtained from livestock and European bison were studied for the presence of tetracycline resistance determinants in their genomes, and the presence of *tet*W and *tet*A(33) was reported in 43.0% and 8.8% of the isolates, respectively, and the investigators concluded that wild ruminants can be a reservoir of tetracycline resistant *T. pyogenes* [96]. On the other hand, seven genomes in the current study harbored an *erm*X gene that protects ribosome inactivation and provides resistance to antibiotics macrolide, lincosamide, and streptogramin. Notably, eight out of the eleven investigated Chinese *T. pyogenes* strains that harbored ARGs were isolated from the pig samples. The presence of these ARGs may be attributed to intensive, large scale swine production wherein indiscriminate use of antimicrobials has become an integral part for not only treating critically ill animals but also for prophylaxis as well as for growth promotion. The high-density swine farms aggravate the hazard of rapid spread of infectious diseases [97]. Moreover, the increased availability of antimicrobials and lower awareness of pig farmers regarding risk and repercussions of antimicrobials and AMR in China is also one of the contributing factors for augmented prevalence [97]. Additionally, the ongoing globalization for trade and travel has also partly increased the dissemination of resistance determinants across the world [98]. Notably, no ARG was detected by CARD in the genomes of DSM20630, NCTC5224, Bu5, and UFV1. The antibiotic resistance profile of the currently investigated genomes along with literature reports that show increased penetration of resistant genes in the *T. pyogenes* strains calls for prudent use of antibiotics in veterinary medicine. These findings emphasize the urgent need to explore novel drug targets and improved vaccines for *T. pyogenes* infection prevention and control. Because human health and livestock productivity is significantly associated with increased antimicrobial resistance, this makes it a global health concern [99]. Therefore, the usage of antimicrobials should be minimized and continuous monitoring of AMR resistance from the national level to a farm level should be carried out to follow the trend, so that veterinarians can choose the most optimal treatment [99,100].

The differential distribution of GI’s, prophages, and ARGs in the investigated strains depicts intraspecific diversity amongst the *T. pyogenes* species and its ability to acquire exogenous DNA in response to various stresses encountered by the bacterium in order to survive inside different niche and hosts.

The study has unravelled an open pan-genome state of *T. pyogenes* and a range of candidate virulence genes, GIs, prophages, and ARGs in the investigated strains. To the best of our knowledge, this is the first descriptive comparative study of *T. pyogenes* genomes which can be used as a starting point for improved understanding of *T. pyogenes* pathogenesis, screening of drug and vaccine targets as well as for diagnostics. Nevertheless, we also emphasize the need of inclusion of greater number of completely sequenced *T. pyogenes* genomes from India as well as from across the world, as well as isolates from different animal hosts for expansion of the present investigation for the understanding of genetic diversity inherent in this opportunistic pathogen, which could enable us in better management of this emerging infectious pathogen.

## 5. Conclusions

*Trueperella pyogenes* is a significant opportunistic bacterium which affects domestic as well as wild animals, and rarely humans by causing pyogenic infections including metritis, mastitis, and pneumonia, which leads to silent economic losses. The increased antimicrobial drug resistance and lack of vaccine with a high degree of protection for infection prevention, and control necessitates detailed genome analysis of the available strains. Genome comparison is an indispensable tool for selection of promising gene target for reverse vaccinology. However, the *T. pyogenes* genomes have not been comprehensively analyzed apart from a few studies that focussed on genomic features of only few virulence determinants. The current investigation involved application of a pan-genomic approach on 19 *T. pyogenes* genomes, which also included an Indian water buffalo isolate Bu5. The investigation revealed a pan-genome repertoire of 3214 CDS, core-genome of 1520 CDS, indispensible genome of 1093, and singletons in the range of 2–63. The phylogenetic analysis utilizing the core-genome showed genetic variability among the analyzed strains with association of clustering of strains with animal hosts. Analysis of known candidate virulence genes such as pyolysin (*plo*), fimbriae (*fimA*, *fimC*, *fimE*,and *fimJ*), collagen-binding protein A (*cbpA*), and neuraminidases (*nanH* and *nanP*) revealed *plo* to be highly conserved amongst them and genetic variability was noted in the rest of the analyzed genes. Further analysis into virulome, mobiliome, and resistome of investigated strains revealed differential distribution and diversity in candidate virulence genes, genomic islands, prophages, and antimicrobial resistance genes. The identified catalogue of candidate virulence genes could be investigated for a nuanced understanding of *T. pyogenes* pathogenesis. Moreover, the identified differences can be further utilized for typing as discriminatory markers for epidemiological surveillance and tracking. In conclusion, the investigation has provided an insight into the genomic repertoire of *T. pyogenes* which can serve as a starting point for future studies related to drug targeting, bacterial typing, diagnostics as well as surveillance purposes.

## Figures and Tables

**Figure 1 antibiotics-12-00024-f001:**
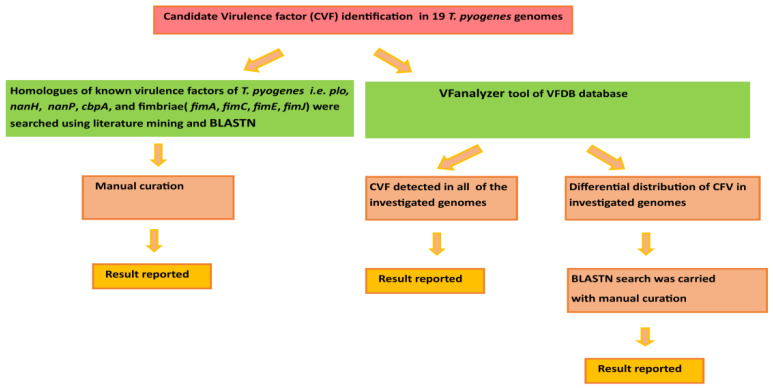
Overview of methodology implemented in identification of candidate virulence factors (CVF’s) in nineteen investigated *T. pyogenes* genomes. The first approach involved the detection of homologues of known virulence factors i.e., pyolysin (*plo*), collagen-binding protein A (*cbpA*), neuraminidases (*nanH* and *nanP*), and fimbriae (*fimA*, *fimC*, *fimE,* and *fimJ*) in the investigated genomes via using literature mining, BLASTN searches, and manual curation. The second approach involved utilization of the VFanalyzer tool of the VFDB database to detect homologues belonging to various classes of virulence factors.

**Figure 2 antibiotics-12-00024-f002:**
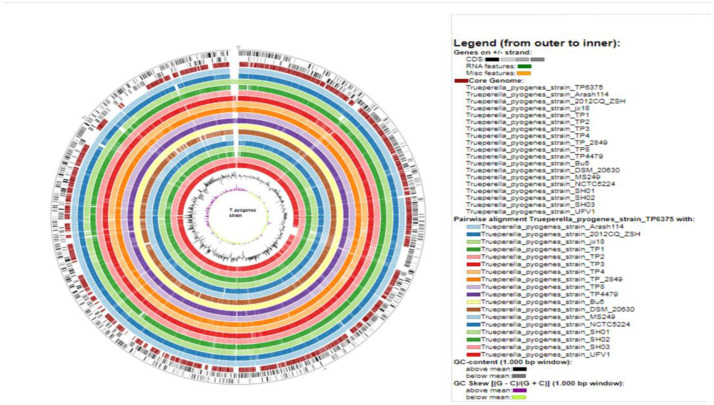
Circular representation of nineteen *T. pyogenes* genomes with reference genome as *T. pyogenes* TP6375.From outside to inside, the figure depicts CDS, core-genome, pairwise alignment of investigated *T. pyogenes* strains against reference genome *T. pyogenes* TP6375, GC content, and GC skew in different colors.

**Figure 3 antibiotics-12-00024-f003:**
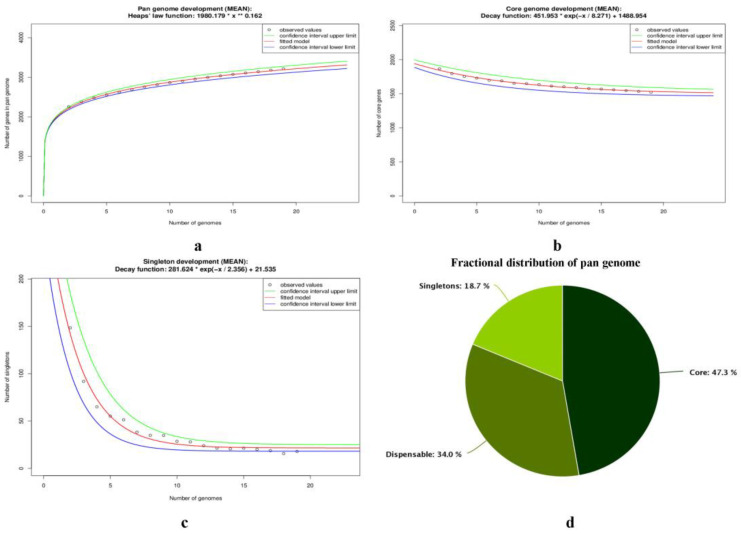
(**a**) The pan-genome development plot projections of nineteen investigated *T. pyogenes* genomes. The red curve illustrates the fitted exponential Heaps’ law function; blue and green curves represent the upper and lower boundary of the 95% confidence interval; (**b**) the core-genome development projections of studied *T. pyogenes* genomes. The red curve represents the fitted exponential decay function; blue and green curves depict the upper and lower boundary of the 95% confidence interval; (**c**) singleton development plot projections of examined *T. pyogenes* genomes; (**d**) fractional distribution of pan-genome of nineteen investigated *T. pyogenes* genomes. * denotes multiplication operator, ** denotes exponentiation operator.

**Figure 4 antibiotics-12-00024-f004:**
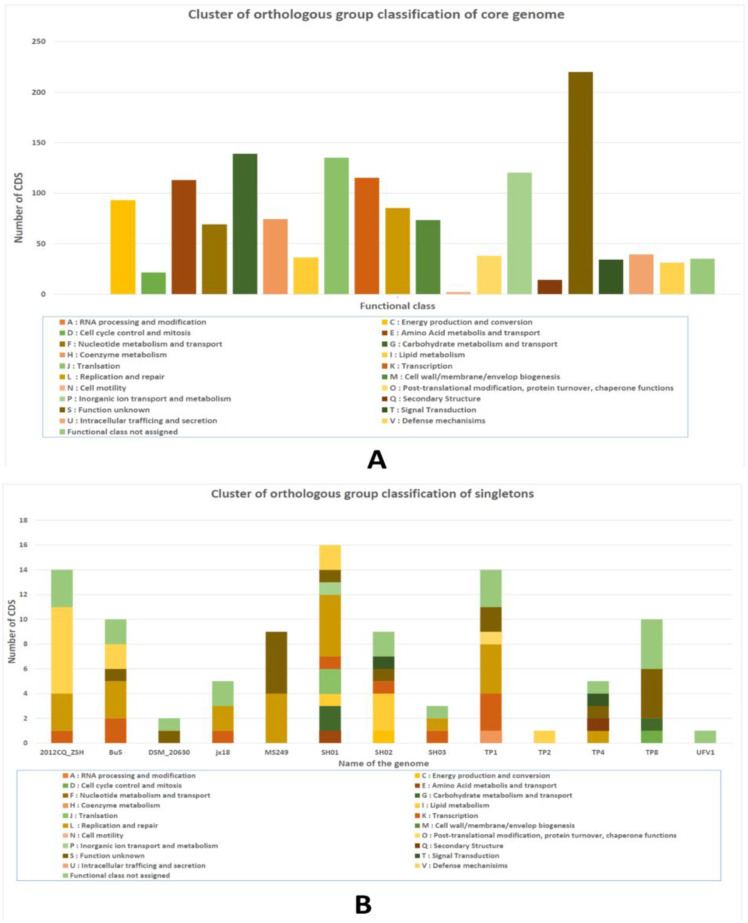
(**A**) COG functional classification of a representative core genome of nineteen investigated *T. pyogenes* genomes as obtained by eggNOG-mapper v2. The *x*-axis denotes the number of CDS, and the *y*-axis represents functional classes. The maximum number of CDS fell in the function unknown category (220 CDS) followed by carbohydrate metabolism and transport category (139 CDS); (**B**) COG functional classification of singletons detected in the investigated *T. pyogenes* genomes by eggNOG-mapper v2.97 CDS (31.29%) out of the total 310 CDS of a complete subset of singletons were queried by the eggNOG-mapper v2. The *x*-axis denotes the number of CDS, and the *y*-axis represents the name of the genome.

**Figure 5 antibiotics-12-00024-f005:**
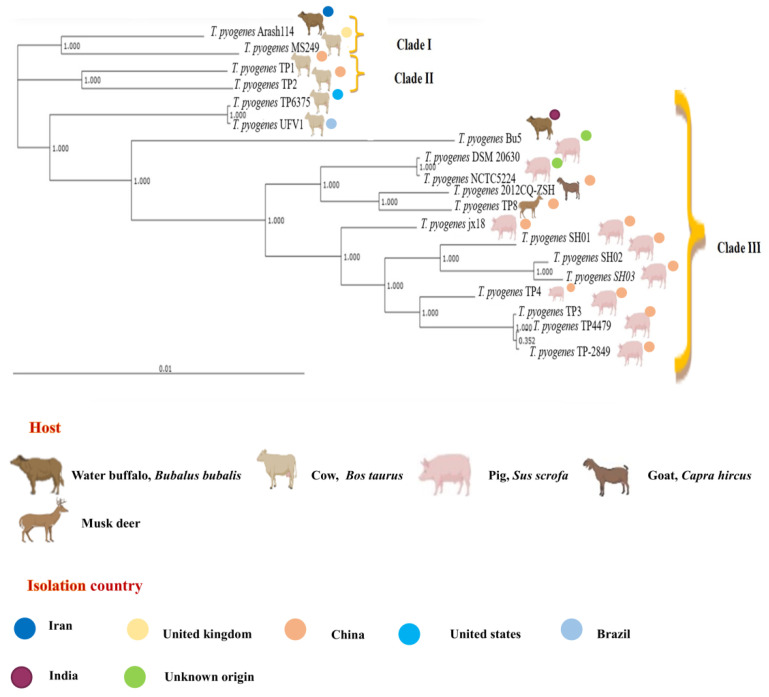
Phylogenetic tree constructed on the basis of core-genome of 19 investigated *T. pyogenes* genomes by EDGAR. The hosts associated with the strains and isolation countries are depicted. The phylogenetic tree can be divided into three clades i.e., clade I, cladeII, and clade III, wherein clade III is harboring the maximum number (11) of strains. The majority of the internal branches have maximum branch support value.

**Figure 6 antibiotics-12-00024-f006:**
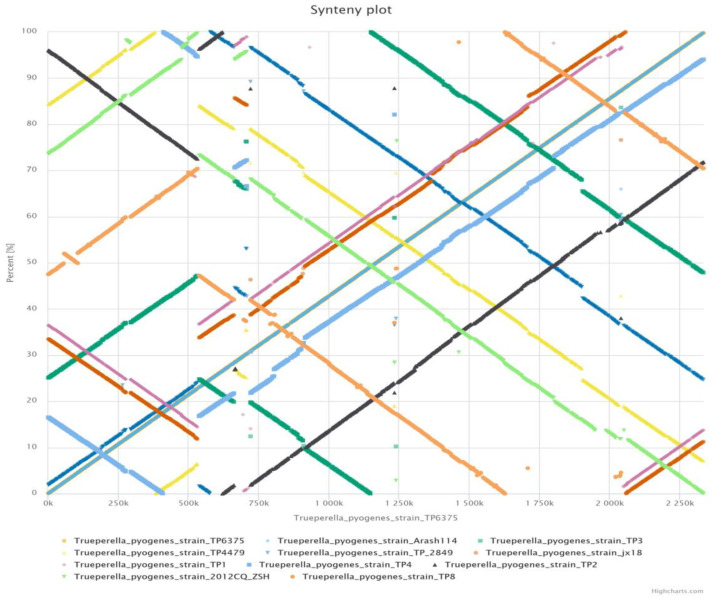
Synteny plot visualization of 10 complete *T. pyogenes* genomes compared against reference genome *T. pyogenes* TP6375 by the EDGAR webserver.

**Figure 7 antibiotics-12-00024-f007:**
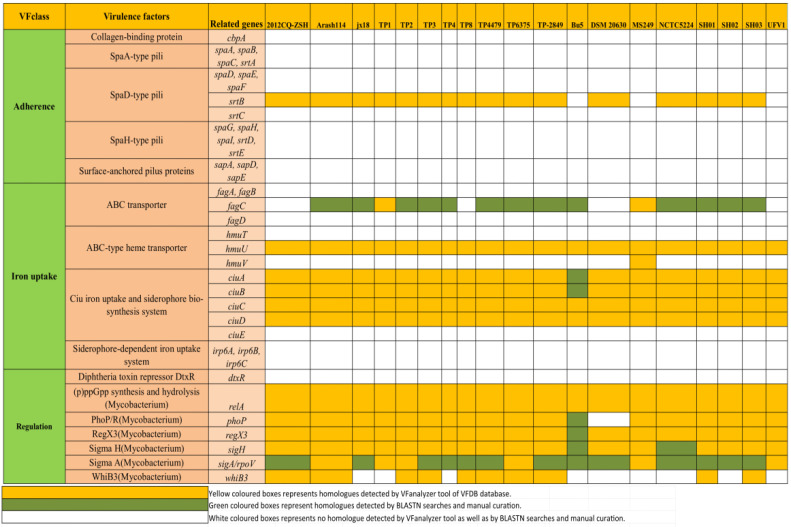
Presence/absence profile of virulence genes homologues aka candidate virulence factors (CVF’s) across the *T. pyogenes* genomes by utilization of the VFanalyzer tool of the VFDB database, BLASTN searches, and manual curation. Yellow colored boxes represent CVF detected by the VFanalyzer tool of the VFDB database. Green colored boxes represent homologues detected by BLASTN searches and manual curation. White colored boxes represent no homologue detected by VFanalyzer tool or BLASTN search.

**Figure 8 antibiotics-12-00024-f008:**
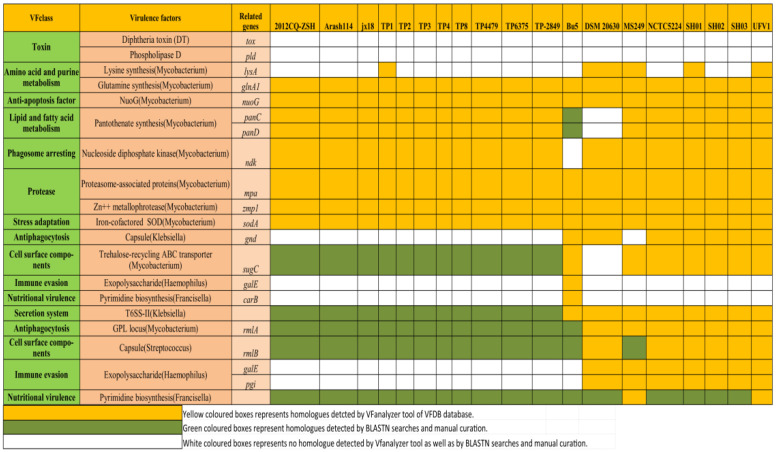
Distribution of CVFs in the nineteen investigated *T. pyogenes* strains belonging to functional classes of toxin, amino acid and purine metabolism, anti-apoptosis factor, lipid and fatty acid metabolism, phagosome arresting, protease, and stress adaptation, anti-phagocytosis, cell surface components, immune evasion, nutritional virulence, and secretion system.

**Figure 9 antibiotics-12-00024-f009:**
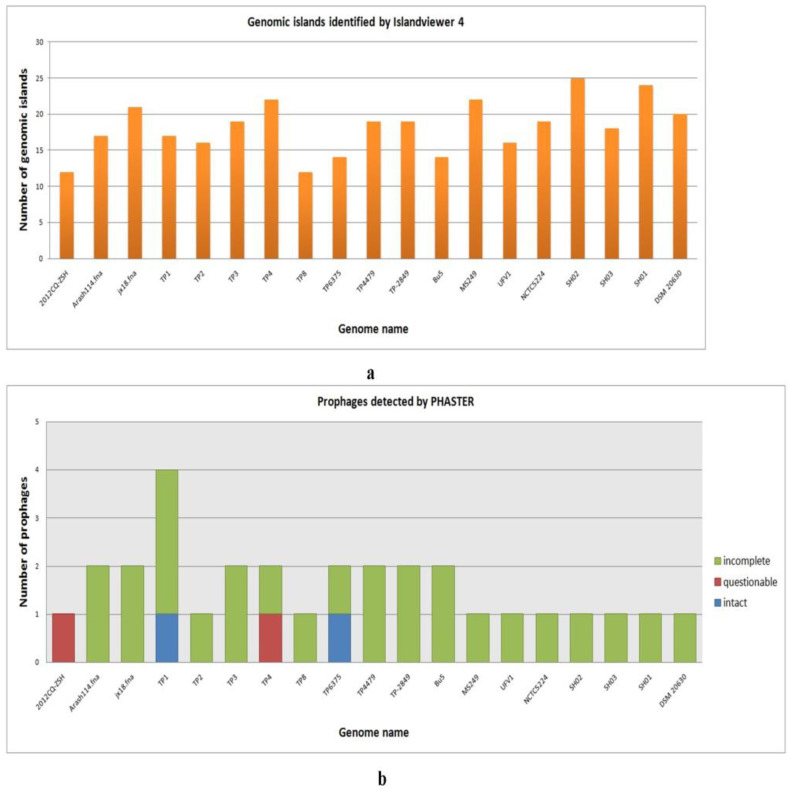
(**a**) Prevalence of genomic islands (GIs) in the investigated 19 *T. pyogenes* genomesas detected by Islandviewer4.The strains harbored GIs in the range of 12–25 with the maximum observed in strain SH01; (**b**) number of prophages detected by PHASTER in the investigated 19 *T. pyogenes* strains. The identified prophages were further classified into intact, incomplete, and questionable categories as depicted by blue, green, and red colors, respectively. Prophages were harbored in the range of 1–4 in the investigated strains with maximum prophages being observed in *T. pyogenes* strain TP1.

**Figure 10 antibiotics-12-00024-f010:**
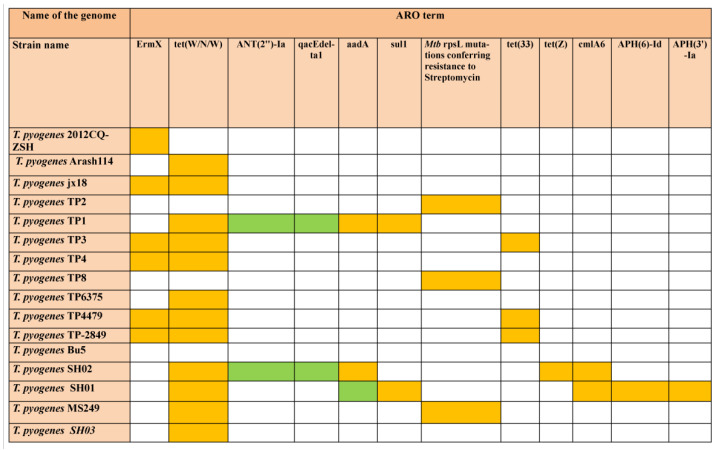
Antimicrobial resistance genes (ARGs) detected in the investigated *T. pyogenes* strains by the CARD database. The yellow and green boxes in the table depict ARGs classified into perfect and strict categories, respectively, by the RGI software of CARD database.

**Table 1 antibiotics-12-00024-t001:** Basic genomic features of the investigated *T. pyogenes* genomes such as isolation country, bases, number of CDS, rRNA, tRNA, tmRNA, and repeat regions.

S.No	Genome and Strain	Host	Country	Bases	GC	CDS	rRNA	tRNA	tmRNA	RR
1	*T. pyogenes* 2012CQ-ZSH	Goat, *Capra hircus*	China	2295822	59.67	2045	6	46	1	-
2	*T. pyogenes* Arash114	Water Buffalo, *Bubalus bubalis*	Iran	2338282	59.49	2109	6	46	1	1
3	*T. pyogenes* jx18	Pig, *Sus scrofa*	China	2415007	59.33	2180	9	46	1	1
4	*T. pyogenes* TP1	Cow, *Bos taurus*	China	2332403	59.76	2126	9	46	1	1
5	*T. pyogenes* TP2	Cow, *Bos taurus*	China	2245225	59.68	1993	9	46	1	1
6	*T. pyogenes* TP3	Pig, *Sus scrofa*	China	2384650	59.35	2112	9	46	1	1
7	*T. pyogenes* TP4	Pig, *Sus scrofa*	China	2427168	59.43	2169	9	47	1	1
8	*T. pyogenes* TP8	Musk dear, *Moschus berezovskii*	China	2272494	59.58	2069	3	45	1	1
9	*T. pyogenes* TP6375	Cow, *Bos taurus*	USA	2338390	59.5	2100	6	46	1	1
10	*T. pyogenes* TP4479	Pig, *Sus scrofa*	China	2382253	59.35	2114	9	46	1	1
11	*T. pyogenes* TP-2849	Pig, *Sus scrofa*	China	2384672	59.35	2113	9	46	1	1
12	*T. pyogenes* Bu5	Water Buffalo, *Bubalus bubalis*	India	2218921	59.66	1948	3	46	1	2
13	*T. pyogenes* MS249	Cow, *Bos taurus*	UK	2216617	59.8	1984	3	46	1	10
14	*T. pyogenes* UFV1	Cow, *Bos taurus*	Brazil	2407507	59.75	2149	2	51	1	2
15	*T. pyogenes* NCTC5224	Pig, *Sus scrofa*	-	2310711	59.57	2073	9	48	1	1
16	*T. pyogenes* SH02	Pig, *Sus scrofa*	China	2380432	59.49	2116	5	46	1	1
17	*T. pyogenes* SH03	Pig, *Sus scrofa*	China	2350892	59.58	2079	7	51	1	1
18	*T. pyogenes* SH01	Pig, *Sus scrofa*	China	2334225	59.49	2068	3	46	1	2
19	*T. pyogenes* DSM 20630	Pig, *Sus scrofa*	-	2187257	59.49	1958	9	45	1	1

## Data Availability

Data available in a publicly accessible repository that does not issue DOIs. Publicly available datasets were analyzed in this study. These data can be found in the NCBI genome (https://www.ncbi.nlm.nih.gov/data-hub/genome/?taxon=1661 accessed on 10 April 2021).

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
