# Peer review of "Comparative Genome Analysis of 19 Trueperella pyogenes Strains Originating from Different Animal Species Reveal a Genetically Diverse Open Pan-Genome"

_antibiotics, 2022, doi:10.3390/antibiotics12010024_

Round 1

Reviewer 1 Report

The manuscript describes comparative genomic analysis of 19 T. pyogenes strains - an opportunistic pathogen. The study was conducted using modern and relevant methods and all  publicly available data. The results described in the article may shed light on the pathogenicity factors of T. pyogenes.

I have only remarks on phylogenetic analysis. What method was used for phylogenetic tree rooting? The tree on Fig.5 seems to be rooted improperly: the upper branches are significantly shorter, than lower. Another questionable thong is clusterization of the leaves of the tree (and resulting clusterization of the strains). Evolutionary distances between Clade III leaves are much longer than such of Clade I and Clade II this fact is counter-intuitive. On the lines 427-429 authors comment on the third clade and write "the other subgroup is solely made up of rest of the 12 Chinese origin porcine strains", while on the Figure 5 there are 10 signs of pig, one for goat and one for musk deer what is different from the text.

After revision of phylogenetic analysis, I assume that this valuable article will be ready for publication.

Author Response

REVIEWER 1

The manuscript describes comparative genomic analysis of 19 T. pyogenes strains - an opportunistic pathogen. The study was conducted using modern and relevant methods and all publicly available data. The results described in the article may shed light on the pathogenicity factors of T. pyogenes.

I have only remarks on phylogenetic analysis. What method was used for phylogenetic tree rooting? The tree on Fig.5 seems to be rooted improperly: the upper branches are significantly shorter, than lower.

Another questionable thong is clusterization of the leaves of the tree (and resulting clusterization of the strains). Evolutionary distances between Clade III leaves are much longer than such of Clade I and Clade II this fact is counter-intuitive.

On the lines 427-429 authors comment on the third clade and write "the other subgroup is solely made up of rest of the 12 Chinese origin porcine strains", while on the Figure 5 there are 10 signs of pig, one for goat and one for musk deer what is different from the text.

After revision of phylogenetic analysis, I assume that this valuable article will be ready for publication.

Response:

Respected reviewer has highlighted a very important aspect of phylogeny, i.e., rooting of tree. Infact, had there been a higher enough number of strains, investigation on common ancestor of the TUs used in the tree would have been more interesting. Since a). The taxa used in the phylogeny are all within species strains of T. pyogenes and b). There are 19 sequences (only) in total; hence the tree has been left un-rooted. Moreover, we have not addressed the query of evolution in the phylogeny, as we focused on the relationship among the strains available. The EDGAR program gives un-rooted tree.

In-spite of this, seeing the distribution of topology of the tree it can be reasonably estimated that root must lay somewhere near base of clade I and II strains. In addition, we do not have (confirmed) distantly related group sequences, which would have helped us in using a methodology for rooting.

Infact, un-rooted trees are highly useful in depicting clusters of related sequences, which is of special interest here as we have same species sequences from different animal species and geographical locations.

The reviewer has pointed out that the upper branches (Clade I & II) are significantly shorter, than lower (Clade III). The reason for this may be that probably the clade I and II belong to the ancestral strains. It can be seen that Clade I strains are from near Far East and Europe; Clade II from Russia; and all of these strains origin are in ruminants. Similarly, strain Bu5 is from another location which may thus again also geographically distant from other strains within Clade III, as other strains of Clade III have shorter distance among themselves, and in addition, they are of porcine origin. This phylogenetic clustering thus is in good accordance with the phylogeography and host species association of the porciine strains of Trueperella. The shorter branch length indicating fewer genetic substitutions (within porcine hosts) and longer branches may be indicate divergence from the Clade I and Clade II taxa (ruminant to porcine). Porcine strains are closer to each other with shorter branch length.

As the revered reviewer very well knows, that  we have used FastTree 1.0, which is based on the “minimum-evolution” principle – it tries to find a topology that minimizes the amount of evolution, or the sum of the branch lengths. We have earlier also tried running our file on EDGAR using neighbour Joining which shows the same clustering.

Another notable point is rate of nucleotide substitution per site, as the scale shows at 0.01. If we look at the length of the scale, the branch lengths after all does not seem to be as long as they appear.

Query: On the lines 427-429 authors comment on the third clade and write "the other subgroup is solely made up of rest of the 12 Chinese origin porcine strains", while on the Figure 5 there are 10 signs of pig, one for goat and one for musk deer what is different from the text.

Response: With regard to comment on the lines 427-429, respected reviewer has rightly pointed out our mistake, which has been corrected.

Reviewer 2 Report

In the current study entitled “Comparative Genome analysis of 19 Trueperella pyogenes strains originating from different animal species reveal a genetically diverse open pangenome”, the athors conducted an extensive comparative genomics analysis to highlight the important genetic determinants associated with the virulence and antibiotic resistance of the opportunistic animal pathogen T. pyogenes. It is a valuable study, but some revisions are necessary to further improve the quality of this work before it can be accepted for publication.

Comments:

1. In the methods section, please add the genome assembly procedure.

2. L259-261: please delete

3. L269: an average read length of 424 bp?

4. L273-274: Is it really the first T. pyogenes genome derived from a water buffalo? How about strain Arash114?

5. L274-276: please place these statements under a new subsection called “Data availability” in the methods section.

6. Table 1: Please add the host animal information in this table.

7. Table 2: The core and pan genome columns should be removed.

8. L343: There were four strains without any unique gene. Please revise the sentence for better clarity.

9. For section 3.4 looking at the functional roles of both the core genome and strain-specific genes, I would like to suggest two changes. When performing functional classification of the core genome, please use only the genes from one strain (for example, TP6375) as the representative of each core gene cluster to allow others better understand the core gene functions of T. pyogenes. It is an unusual practice to include all the core genes of every single strain when doing functional annotations. Next, a stacked bar graph should be used when showing the functional roles of the genes unique to each strain. In the current figure, one cannot tell the functions of genes which are exclusive to individual strain. Please revise the figure and text as appropriate.

10. L591: please add a sentence or two describing what antibiotic resistance would each identified gene confer.

11. Another round of proofreading is recommended to correct several grammatical and punctuation errors in the text.

Author Response

Reviewer 2

In the current study entitled “Comparative Genome analysis of 19 Trueperella pyogenes strains originating from different animal species reveal a genetically diverse open pangenome”, the authors conducted an extensive comparative genomics analysis to highlight the important genetic determinants associated with the virulence and antibiotic resistance of the opportunistic animal pathogen T. pyogenes. It is a valuable study, but some revisions are necessary to further improve the quality of this work before it can be accepted for publication.

Comments:

  1. In the methods section, please add the genome assembly procedure.

Response: Same has been added

  1. L259-261: please delete

Response: Lines have been deleted.

  1. L269: an average read length of 424 bp?

Response: Yes sir, as you know, the 454 GS-FLX Titanium technology provides around 1,000,000 sequences in a single 10-hour run. These sequences, with an average read length equal to 330 bp, may be up to 500 bp in shotgun libraries conditions, much longer than can be obtained with the other available approaches. Gilles, A., Meglécz, E., Pech, N. et al. Accuracy and quality assessment of 454 GS-FLX Titanium pyrosequencing. BMC Genomics 12, 245 (2011). https://doi.org/10.1186/1471-2164-12-245).

  1. L273-274: Is it really the first T. pyogenes genome derived from a water buffalo? How about strain Arash114?

Response: Yes, the strain Bu5 is the first genome of Trueperella pyogenes in the world from water buffalo. It was accessioned on 11/11/2017. Arash genome was accessioned in 18/04/2018 (See PATRIC and https://www.ncbi.nlm.nih.gov/genome/browse/#!/prokaryotes/16424/.

  1. L274-276: please place these statements under a new subsection called “Data availability” in the methods section.

Response: These statements have been placed under new subsection “Data availability” as suggested, pl.

  1. Table 1: Please add the host animal information in this table.

Response: As suggested, the host animal information has been added. In order to adjust the contents in the table, some changes have been brought like GC content has now been indicated as GC; Repeat region as RR etc.

  1. Table 2: The core and pan genome columns should be removed.

Response: As suggested the Table 2 has been removed, along with its reference in Text.

  1. L343: There were four strains without any unique gene. Please revise the sentence for better clarity.

Response: As suggested, the sentence has been revised by creating a new separate sentence.

  1. For section 3.4 looking at the functional roles of both the core genome and strain-specific genes, I would like to suggest two changes. When performing functional classification of the core genome, please use only the genes from one strain (for example, TP6375) as the representative of each core gene cluster to allow others better understand the core gene functions of T. pyogenes. It is an unusual practice to include all the core genes of every single strain when doing functional annotations. Next, a stacked bar graph should be used when showing the functional roles of the genes unique to each strain. In the current figure, one cannot tell the functions of genes which are exclusive to individual strain. Please revise the figure and text as appropriate.

Response: As suggested by reviewer, both the changes have been brought about. For core genome, only strain TP6375 has been used. Secondly, when showing the singletons gene functional annotation, a stacked bar graph has been used. The new figure (Fig 4) has been inserted. Relevant changes have been also incorporated in material and methods, results and discussion section related to above changes.

  1. L591: please add a sentence or two describing what antibiotic resistance would each identified gene confer.

Response: As suggested by esteemed reviewer, we have added a sentence highlighting the antimicrobials and the genes (in bracket) conferring resistance.

  1. Another round of proofreading is recommended to correct several grammatical and punctuation errors in the text.

Response: As suggested, another proofreading has been performed to correct the grammatical and/or punctuation errors, pl.

Round 2

Reviewer 2 Report

Thank you for improving the manuscript as suggested. Other than placing the "Data availability" statement under the methods section, I have no further comment and believe that this manuscript is now suitable for publication in Antibiotics.